# Replicating viral vector platform exploits alarmin signals for potent CD8$^+$ T cell-mediated tumour immunotherapy

Sandra M. Kallert[1], Stephanie Darbre[2], Weldy V. Bonilla[1], Mario Kreutzfeldt[2,3], Nicolas Page[2], Philipp Müller[4,†], Matthias Kreuzaler[4], Min Lu[1], Stéphanie Favre[5], Florian Kreppel[6], Max Löhning[7,8], Sanjiv A. Luther[5], Alfred Zippelius[4,9], Doron Merkler[2,3,*] & Daniel D. Pinschewer[1,*]

Viral infections lead to alarmin release and elicit potent cytotoxic effector T lymphocyte (CTL$^{eff}$) responses. Conversely, the induction of protective tumour-specific CTL$^{eff}$ and their recruitment into the tumour remain challenging tasks. Here we show that lymphocytic choriomeningitis virus (LCMV) can be engineered to serve as a replication competent, stably-attenuated immunotherapy vector (artLCMV). artLCMV delivers tumour-associated antigens to dendritic cells for efficient CTL priming. Unlike replication-deficient vectors, artLCMV targets also lymphoid tissue stroma cells expressing the alarmin interleukin-33. By triggering interleukin-33 signals, artLCMV elicits CTL$^{eff}$ responses of higher magnitude and functionality than those induced by replication-deficient vectors. Superior anti-tumour efficacy of artLCMV immunotherapy depends on interleukin-33 signalling, and a massive CTL$^{eff}$ influx triggers an inflammatory conversion of the tumour microenvironment. Our observations suggest that replicating viral delivery systems can release alarmins for improved anti-tumour efficacy. These mechanistic insights may outweigh safety concerns around replicating viral vectors in cancer immunotherapy.

[1] Division of Experimental Virology, Department of Biomedicine, University of Basel, Petersplatz 10, 4009 Basel, Switzerland. [2] Departement de Pathologie et Immunologie, Centre Médical Universitaire, University of Geneva, 1 rue Michel Servet, 1211 Geneva, Switzerland. [3] Division of Clinical Pathology, Geneva University Hospital, Centre Médical Universitaire, 1 rue Michel Servet, 1211 Geneva, Switzerland. [4] Department of Biomedicine, University Hospital and University of Basel, Hebelstr. 20, 4031 Basel, Switzerland. [5] Department of Biochemistry, Center for Immunity and Infection Lausanne, University of Lausanne, Chemin des Boveresses 144, 1066 Epalinges, Switzerland. [6] Witten/Herdecke University (UW/H), Faculty of Health/School of Medicine, Stockumer Str. 10, 58453 Witten, Germany. [7] Experimental Immunology and Osteoarthritis Research, Department of Rheumatology and Clinical Immunology, Charité–Universitätsmedizin Berlin, Charitéplatz 1, 10117 Berlin, Germany. [8] Pitzer Laboratory of Osteoarthritis Research, German Rheumatism Research Center (DRFZ), Leibniz Institute, Charitéplatz 1, 10117 Berlin, Germany. [9] Department of Medical Oncology, University Hospital Basel, Hebelstr. 20, 4031 Basel, Switzerland. † Present address: Department of Cancer Immunology and Immune Modulation, Boehringer Ingelheim Pharma GmbH & Co. KG, Birkendorfer Str. 65, 88400 Biberach an der Riss, Germany. * These authors contributed equally to this work. Correspondence and requests for materials should be addressed to D.D.P. (email: daniel.pinschewer@unibas.ch).

The clinical efficacy of checkpoint blockade, oncolytic viruses and adoptive T-cell therapy heralds success in harnessing the immune system in the combat against cancer[1–3]. Conversely, active immunization has not yet demonstrated consistent efficacy in clinical Phase III trials[4–6], raising an urgent need for improved vaccine formulations that should aim at delivering large numbers of $CD8^+$ cytotoxic effector T lymphocytes ($CTL^{eff}$) to the tumour site while simultaneously establishing a pool of self-replenishing memory cells for durable tumour control[7–9]. The immunological response profile of replicating viral vaccines represents an excellent match to these requirements[10]. By delivering tumour-associated antigens (TAA) in the context of an acute viral infection, such delivery systems should supply critical alarmin signals, also referred to as damage-associated molecular patterns, as well as pathogen-associated molecular patterns (PAMPs) for optimal CTL induction and differentiation[11,12].

Classical tumour vaccination regimens such as peptides-in-adjuvant showed only marginal clinical benefit, despite the induction of sizeable tumour antigen-specific CTL responses[13]. Major impediments include inefficient tumour infiltration and efficacy of specific $CTL^{eff}$ (refs 14,15). Overcoming these hurdles will critically depend on innate immune activation, which can be achieved by virus-induced inflammation[16].

Lymphocytic choriomeningitis virus (LCMV), the prototype member of the arenavirus family, elicits $CTL^{eff}$ responses of high magnitude and cytolytic capacity, in combination with life-long CTL immunity. These features, together with a low hazard profile for humans, have rendered it a primary workhorse of immunologists since the 1930s (ref. 17). Experimental LCMV infections in humans have documented a systemic inflammatory reaction, accompanied by a lymphoblastic reaction in peripheral blood[18], which was reminiscent of the massive $CTL^{eff}$ response in infectious mononucleosis[19]. Accordingly, studies in accidentally LCMV-infected laboratory workers have confirmed that, analogously to mice, high frequencies of effector memory CTL are maintained for several years after a single acute infection[20]. These features in combination with low seroprevalence in the human population[21,22] raised our interest in LCMV as a live-attenuated cancer immunotherapy platform, to deliver TAA-specific immunization alongside with potent innate immune activation.

We and others have previously reported that interleukin-33 (IL-33) is a key driver of potent and protective $CTL^{eff}$ responses to several replicating RNA and DNA viruses[11]. Subsequently, IL-33 was found to also be essential for antiviral Th1 CD4 responses[23] and for graft-versus-host disease[24]. Collectively, these reports underscore the global significance of IL-33 in promoting type 1 immune responses (reviewed in ref. 25), besides an additional undisputed function in type 2 immune responses such as in the context of allergy and immunity to parasites. Upon release from necrotic cells, IL-33 signals through its receptor ST2 on activated $CD8^+$ T cells, thereby enhancing clonal expansion, effector differentiation and ultimately CTL efficacy[11]. By contrast to wild-type LCMV (LCMVwt), however, replication-deficient LCMV-based vectors (rLCMV[26]) fail to trigger the IL-33 alarmin pathway[11].

Hence we engineer a TAA delivery platform based on live-attenuated LCMV (artLCMV). Unlike replication-deficient rLCMV vectors, artLCMV targets not only dendritic but also lymphoid stromal cells, thereby triggering the IL-33–ST2 alarmin pathway and inducing superior TAA-specific $CTL^{eff}$ responses. The activation of critical innate immune pathways including alarmins represents a discriminating feature of replicating microbial delivery systems, which might be decisive for the success of active cancer immunotherapy.

## Results

**Generation of genetically stable and live-attenuated artLCMV.** We aimed at combining tumour antigen vaccination with infection-induced alarmin signals. Hence we sought a strategy how to stably incorporate transgenes into replicating LCMV. The virus' genome consists of two negative-stranded RNA segments designated L and S, respectively, encoding two viral genes each (LCMVwt, Fig. 1a). 5' and 3' untranslated regions (UTR) and an intergenic region (IGR) flank the ORFs on each segment (Fig. 1a). Reverse genetic tools are available to efficiently tailor the infectious virus' genome for medical application[26,27]. As reported[28], transgenes of interest can be accommodated in the LCMV genome by segregating the viral glycoprotein (GPC) and nucleoprotein (NP) genes onto artificially duplicated S segments ($S_{NP}$, $S_{GP}$ in r3LCMV, Fig. 1a). According to the originally published vector design strategy[28], both NP and GPC remain under control of their respective regulatory RNA elements (natural positioning; r3LCMV). Genetic and phenotypic stability represent key criteria for manufacturing and clinical translation of live-attenuated viral vector systems[29,30]. Accordingly, observations on transgene loss in r3LCMV-infected animals (see below) prompted us to search for molecular strategies to stabilize r3LCMV genomes. By placing GPC under 3'UTR control, we generated viruses with an artificial genome organization (artLCMV, Fig. 1a). The reasoning was that, although a rare event in negative strand RNA viruses[31], inter-segmental recombination[32] would reunite NP and GPC on a single S segment, thereby deleting the transgenes (Fig. 1b). The arenavirus promoter consists of an intra-segmental RNA hybrid formed by a highly conserved stretch of 19–21 terminal nucleotides in each segment's 5'UTR and 3'UTR. Therefore, viral promoter activity requires both, a 5'UTR and a 3'UTR on each RNA template (Fig. 1b)[33]. Unlike in r3LCMV, a recombination event reuniting the NP and GPC ORFs of artLCMV would occur at the expense of losing the 5'UTR, thus creating an inactive recombination product devoid of a viral promoter (Fig. 1c). Both, r3LCMV and artLCMV were significantly attenuated in cell culture (Fig. 1d). Flow cytometric studies with green (GFP) and red fluorescent protein (tomato) co-expressing viruses (r3LCMV-GFP/tom, artLCMV-GFP/tom) revealed that red/green double-positive virions were outnumbered by a >tenfold excess of bi-segmented replication-deficient particles, carrying either only $S_{GP}$ or $S_{NP}$ in combination with the L segment (Fig. 1e, see Supplementary Fig. 1 for flow cytometry gating strategies). An immunofocus assay (IFF)-based quantification of $S_{NP}$-only and $S_{GP}$-only virions yielded analogous results (Supplementary Fig. 2a). Hence, attenuation was, at least in part, due to inefficient co-packaging of $S_{NP}$ and $S_{GP}$ segments. To study the genetic stability of artLCMV and r3LCMV we exploited AGR mice. Owing to targeted deletions of RAG1 (T-cell and B-cell deficiency) as well as of the type I and type II interferon genes, AGR mice readily reveal an attenuated virus' reversion to virulence[34]. We measured viral loads in blood by IFF, relying on the detection of the viral structural protein NP (NP-IFF, Fig. 2a). Within the first 20 days after infection, r3LCMV and artLCMV remained at considerably lower titres in blood than LCMVwt, which was in line with published data documenting in vivo attenuation of r3LCMV[35]. After 30–40 days, however, r3LCMV-infected AGR mice reached NP-IFF titres identical to those infected with LCMVwt. Conversely, artLCMV viremia remained at low levels throughout the observation period of 120 days (Fig. 2a). In artLCMV-infected animals, the assessment of total infectivity by NP-IFF and the quantification of GFP-expressing viruses by GFP-IFF yielded comparable titres on day 5 as well as on day 120 after infection, indicating that the transgene had been retained (Fig. 2b,c). In contrast,

r3LCMV-GFP-infected mice turned GFP-IFF-negative while abundant NP-IFF infectivity persisted, suggesting that the virus had lost its transgene. Indeed, the viral NP was abundant in the liver and spleen of r3LCMV-GFP-infected mice, as expected, while GFP was absent (Fig. 2d, Supplementary Fig. 2b). In contrast, liver and spleen of artLCMV-GFP-infected animals exhibited dense green fluorescence (Fig. 2d, Supplementary Fig. 2b; analogous data for blood monocytes in Supplementary Fig. 2c). To verify whether r3LCMV-GFP had inactivated its transgene by recombining its two S segments, as hypothesized (Fig. 1b), we relied on an PCR with reverse transcription (RT-PCR)-based sequencing strategy (Supplementary Fig. 2d). Recombined S segment RNA species ($S_{rec}$, Supplementary Fig. 2d) reuniting $NP$ and $GPC$ sequences in a single RNA molecule were detected in ten out of ten r3LCMV-GFP-infected AGR mice from two independent series of experiments. In contrast, the blood of 11 artLCMV-infected mice was free of detectable $S_{rec}$ RNA.

Sequence analysis of $S_{rec}$ segments revealed that individual non-homologous recombination events had occurred in each animal, with breakpoints in the $GFP$–IGR border region (Fig. 1b, Supplementary Fig. 2e). Genetic tags in the IGR and $GPC$ sequences of r3LCMV-GFP excluded laboratory contaminations as potential confounder in these analyses and assigned genetic elements in $S_{rec}$ to its parental $S_{NP}$ or $S_{GP}$ segments (Supplementary Fig. 2f,g). We found that a cDNA-derived bi-segmented virus with an exemplary $S_{rec}$ segment ($S_{rec}$#1) grew to LCMVwt titres in cell culture (Fig. 2e). In combination with the inefficient co-packaging of $S_{NP}$ and $S_{GP}$ in tri-segmented particles (Fig. 1d), this finding explained the selective advantage and consistent outgrowth of $S_{rec}$ segments in r3LCMV-infected AGR mice.

**Apathogenic artLCMV induces potent CTL[eff] responses.** *In vivo* attenuation represents a prerequisite for the use of live vaccine delivery platforms in immunocompromised cancer patients.

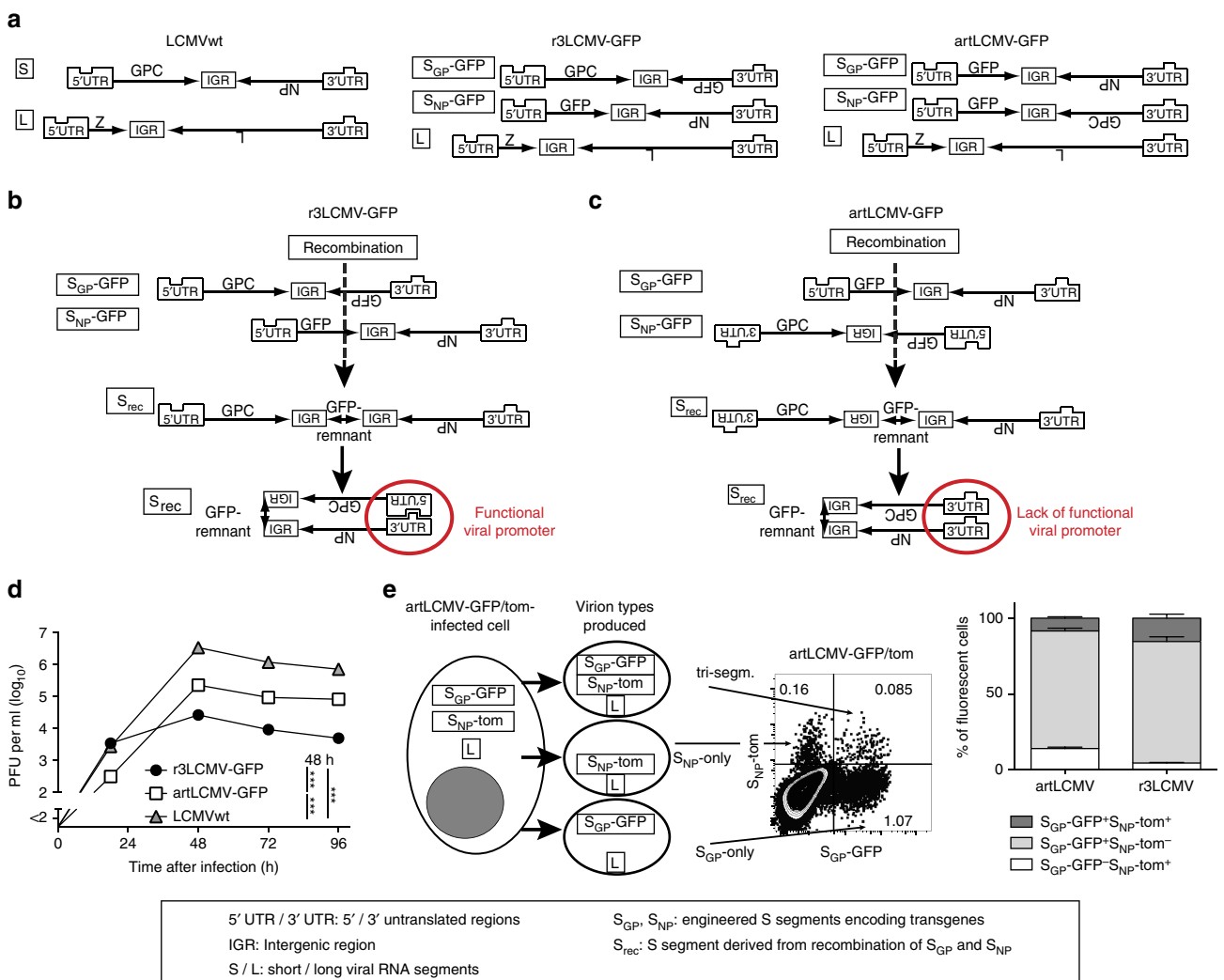

**Figure 1 | Genetic design and growth of artLCMV in cell culture.** (**a**) Genome organization of LCMVwt, r3LCMV-GFP and artLCMV-GFP. (**b**) Non-homologous recombination unites $GPC$ and $NP$ ORFs of r3LCMV-GFP yielding $S_{rec}$ with only $GFP$ remnants. 3′UTR and 5′UTRs of $S_{rec}$ pair to form a functional promoter (Supplementary Fig. 2). (**c**) Hypothetical recombination event uniting $GPC$ and $NP$ of artLCMV-GFP at the expense of losing the 5′UTR. Duplicated 3′UTRs on such hypothetical $S_{rec}$ cannot form a functional promoter. (**d**) Growth kinetics of LCMVwt, artLCMV-GFP and r3LCMV-GFP on BHK-21 cells. Symbols represent the mean of three replicates (s.e.m. hidden in symbol size). PFU: plaque forming units. $N = 3$ ($N$: number of independent data sets). (**e**) artLCMV-GFP/tom and r3LCMV-GFP/tom (containing $S_{GP}$-GFP and $S_{NP}$-tom) contained tri-segmented and bi-segmented particles, which were discriminated and quantified by FACS analysis 12 h after infection of BHK-NP cells. Representative FACS plot for artLCMV-GFP/tom (analogous results for r3LCMV-GFP/tom, see quantification in bar graph). Bars represent the mean + s.e.m. of three replicates per group. $N = 3$. Data in **d** (48 h after infection) were analysed by one-way analysis of variance (ANOVA) with Bonferroni *post hoc* test. ***$P < 0.001$.

Infection of wt mice with LCMVwt, but not with artLCMV resulted in viremia (Fig. 3a). Moreover, low level artLCMV infectivity was only transiently detected in spleen and liver on day 4 and was cleared by day 7, whereas LCMVwt persisted for 10 days at high titres (Fig. 3b). LCMVwt can cause choriomeningitis in accidentally infected humans and in mice the virus is invariably lethal at intracranial (i.c.) doses of $\leq 10$ plaque forming units (PFU, Fig. 3c, ref. 36). In contrast, artLCMV failed to cause disease at doses up to $10^4$ PFU i.c., and only one of five animals developed terminal disease when given $10^5$ PFU i.c. Despite its attenuation, intravenous artLCMV-induced substantial levels of serum type I interferon (IFN-I, Fig. 3d) for at least 48 h. rLCMV triggered a comparably minor and short-lived IFN-I release, whereas neither rAd-based nor replicating vaccinia virus-based vectors induced detectable levels of systemic IFN-I. Analogous results were obtained when rAd vectors were given intramuscularly or upon subcutaneous and intradermal administration of vaccinia virus-based vectors (Supplementary Fig. 3a). Although we used replication-competent vaccinia virus for our study, the lack of systemic IFN-I was expected owing to multiple virally encoded antagonists of the IFN-I response[37]. Taken together, these results indicated that artLCMV was genetically stable and substantially attenuated, both in vitro and in vivo, but retained the ability to efficiently activate the innate immune system. To assess artLCMV-induced CD8$^+$ T-cell responses to transgenes of choice, we generated an ovalbumin- (OVA-) expressing vector (artLCMV-OVA). It induced OVA epitope-specific CTL numbers, which were substantially higher than those elicited by an OVA-recombinant adenovirus 5-based vector (rAd-OVA) or by a replication-deficient LCMV-based vector (rLCMV-OVA, Fig. 4a). Most notably, artLCMV-OVA-triggered KLRG1$^+$CD127$^-$ effector CD8$+$ T-cell (CTL$^{eff}$) responses in blood and spleen

exceeded those of the other delivery systems by $\sim$tenfold (Fig. 4b,c), while CD127$^+$ CTL$^{mem}$ memory precursor cell numbers were comparable (Fig. 4d). OVA-specific total CTL and CTL$^{eff}$ responses to artLCMV-OVA immunization were also significantly higher than those induced upon intramuscular administration of rAd-OVA and either subcutaneous or intradermal vaccination with rVACC-OVA (Supplementary Fig. 3b,c). Intracellular cytokine staining showed that artLCMV-OVA elicited ten to 20-fold higher numbers of IFN-$\gamma$-producing and IFN-$\gamma$/TNF co-producing CD8$^+$ T cells than rAd-OVA or rLCMV-OVA, and also IFN-$\gamma$/TNF/IL-2 triple producers were significantly more numerous (Fig. 4e). These differences in magnitude and functionality of OVA-specific CTL populations persisted in the memory phase (Fig. 4f). Superior functionality of artLCMV-induced CTL$^{eff}$ responses was also evident in primary ex vivo Cr$^{51}$ release assays, with $\geq$tenfold more lytic units in spleen after artLCMV infection than after rLCMV-OVA, rAd-OVA or recombinant vaccinia virus (rVACC-OVA) immunization (Fig. 4g). Importantly, artLCMV-based vaccines elicited also robust CD8$^+$ T-cell responses against the tumour self-antigens Her2 and P1A (Fig. 4h,i). Analogously to artLCMV-OVA immunization, P1A-specific CD8$^+$ T cells in blood of artLCMV-P1A-immunized mice were $\sim$tenfold more numerous than in animals receiving rLCMV-P1A (Fig. 4i). LCMVwt evades antibody neutralization by means of its envelope glycan shield[38], and even two sequential artLCMV immunizations failed to induce detectable neutralizing serum activity (Fig. 4j). Accordingly, artLCMV-induced CTL responses were considerably augmented when artLCMV-OVA was re-administered in homologous prime-boost vaccination (Fig. 4k). These observations do not, however, rule out the likely possibility that alternative mechanisms of interference

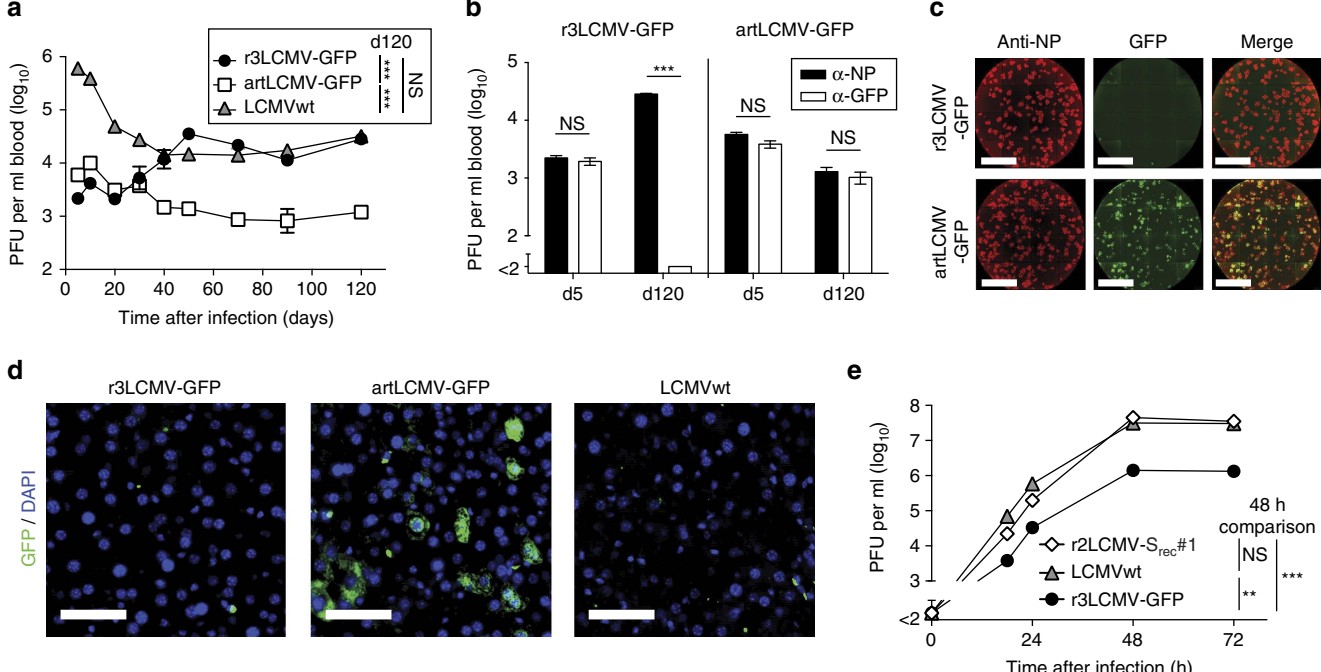

**Figure 2 | Stable transgene expression and attenuation of artLCMV in vivo.** (**a–d**) Viremia of AGRAG mice infected with r3LCMV-GFP ($n = 7$), artLCMV-GFP ($n = 7$) or LCMVwt ($n = 3$, **a,d**). Symbols show the mean ± s.e.m. $N = 2$. Blood samples obtained on d5 and d120 were processed for NP-IFF and GFP-IFF (**b**). Bars represent the mean ± s.e.m. of seven mice. $N = 2$. Representative NP-IFF and GFP-IFF analyses of d150 serum (**c**, scale bar, 0.5 cm). Representative liver sections analysed for GFP$^+$ cells on d150 (**d**, scale bar, 50 μm). (**e**) Growth kinetics of LCMVwt, r3LCMV-GFP and r2LCMV-S$_{rec}$#1 on BHK-21 cells (compare Supplementary Fig. 2e). Symbols represent the mean of three replicates (s.e.m. error bars hidden in the symbol size). $N = 2$. Data in **a** (120 days after infection) and **e** (48 h after infection) were analysed by one-way ANOVA with Bonferroni *post hoc* test. Data in **b** were analysed by unpaired two-tailed Student's *t*-test. NS, not significant; **$P < 0.01$ and ***$P < 0.001$.

such as T-cell immunity to viral backbone epitopes[39] may compete with and thereby attenuate booster responses to artLCMV-vectorized transgenes.

**artLCMV triggers IL-33-driven CTL by targeting stromal cells.** We investigated the mechanisms underlying exceptionally potent CTL[eff] responses upon artLCMV immunization. First we studied its tropism for antigen-presenting cells *in vivo* as a basis for potent CTL induction. While both, rLCMV and artLCMV, efficiently targeted plasmacytoid DCs (pDCs), artLCMV infected a significantly higher number of conventional dendritic cells and macrophages than its replication-deficient counterpart rLCMV (Fig. 5a). In addition we hypothesized that artLCMV, analogously to LCMVwt, triggered the IL-33–ST2 alarmin pathway, thereby potentiating CTL responses and CTL[eff] responses in particular[11]. When immunized with artLCMV-OVA, wt mice mounted ~tenfold higher OVA-specific CTL responses than animals lacking the IL-33 receptor ST2 (*Il1rl1*[−/−]), both in spleen and blood (Fig. 5b,c). These differences were particularly pronounced in the KLRG1[+]CD127[−] CTL[eff] subset (Fig. 5b,c). Conversely, the responses to rAd-OVA, rVACC-OVA and rLCMV-OVA were unaffected by ST2 deficiency, suggesting that artLCMV-OVA immunization but neither replication-deficient rLCMV- nor rAd- or vaccinia-vectored vaccination triggered the IL-33–ST2 axis. The capacity to replicate *in vivo* differentiates artLCMV and rLCMV and was apparently required to activate this pathway. Bioactive IL-33 is released from non-haematopoietic stromal cells[11], which are a target of replicating LCMV infection[40]. To test the hypothesis that artLCMV triggered ST2 signalling by infecting IL-33-expressing stromal cells inside secondary lymphoid organs, we developed and validated a green fluorescent IL-33 reporter mouse (*IL-33*[gfp/wt], Supplementary Fig. 4a,b). More than half of the splenic CD45[neg] (non-haematopoietic) gp38[+] fibroblastic reticular cells (FRC) reported IL-33 in naïve animals

(IL-33-GFP[+]), whereas CD31[+] blood endothelial cells (BEC) were virtually IL-33-GFP-negative (Supplementary Fig. 4c). Red fluorescent artLCMV-tom but not rLCMV-tom immunization yielded a distinct population of infected and IL-33-expressing (tomato[+]IL-33-GFP[+]) FRCs, confirming that replicating artLCMV but not replication-deficient rLCMV infected IL-33-expressing FRCs in spleen (Fig. 5d). In addition, both rLCMV and artLCMV infected a sizeable number of splenic BECs (Fig. 5d). A subset of artLCMV-infected BECs also reported IL-33, whereas rLCMV-infected BECs were virtually uniformly IL-33-GFP-negative (Fig. 5d). IL-33-reporting BECs were highly overrepresented amongst those infected by artLCMV (Supplementary Fig. 4d), suggesting that artLCMV infection either had a predilection for a rare subset of IL-33-expressing BECs or that it triggered IL-33 expression in infected BECs. Both artLCMV-infected (tom[+]) IL-33-GFP[+] BEC and FRC frequencies declined sharply between day 3 and 7 after immunization (Fig. 5d). This observation lent support to the hypothesis that bioactive IL-33 was released from dying artLCMV-infected lymphoid tissue stromal cells, offering a mechanism whereby replicating viral delivery systems supply IL-33 to the ensuing CTL response inside secondary lymphoid organs.

**IL-33-driven tumour control upon artLCMV immunotherapy.** To compare the efficacy of several viral vector platforms in cancer immunotherapy, we first exploited a transplantable OVA-expressing tumour model (EG7-OVA). Unlike rLCMV-OVA and rAd-OVA, which showed partial or no clinical benefit, respectively, when administered to mice with an established solid tumour, treatment with artLCMV-OVA afforded substantial tumour control and prolonged the animals' survival in a CD8[+] T-cell-dependent manner (Fig. 6a,b, Supplementary Fig. 5a,b). Importantly, an irrelevant artLCMV vector (artLCMV-GFP)

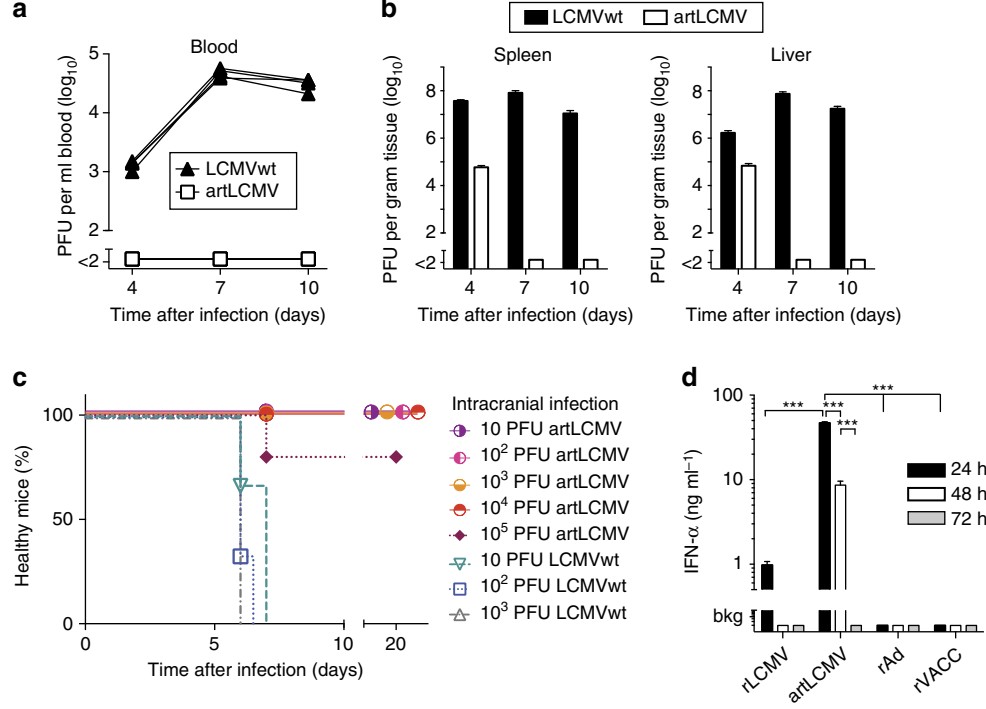

**Figure 3 | artLCMV is attenuated *in vivo* and induces systemic type I interferon.** (**a,b**) Viral loads in blood (**a**), spleen and liver (**b**) of C57BL/6 mice. Symbols and lines show individual mice (overlaid for artLCMV, **a**), bars represent the mean + s.e.m. (**b**) (*n* = 4). *N* = 2. (**c**) Choriomeningitis incidence in i.c. infected mice. Terminally diseased animals were euthanized. (*n* = 3–5). (**d**) Serum IFN-α after artLCMV, rLCMV, rAd or rVACC immunization. Bars represent the mean + s.e.m. of four mice. *N* = 2. Data in **d** were analysed by two-way ANOVA with Bonferroni *post hoc* test. ***P < 0.001.

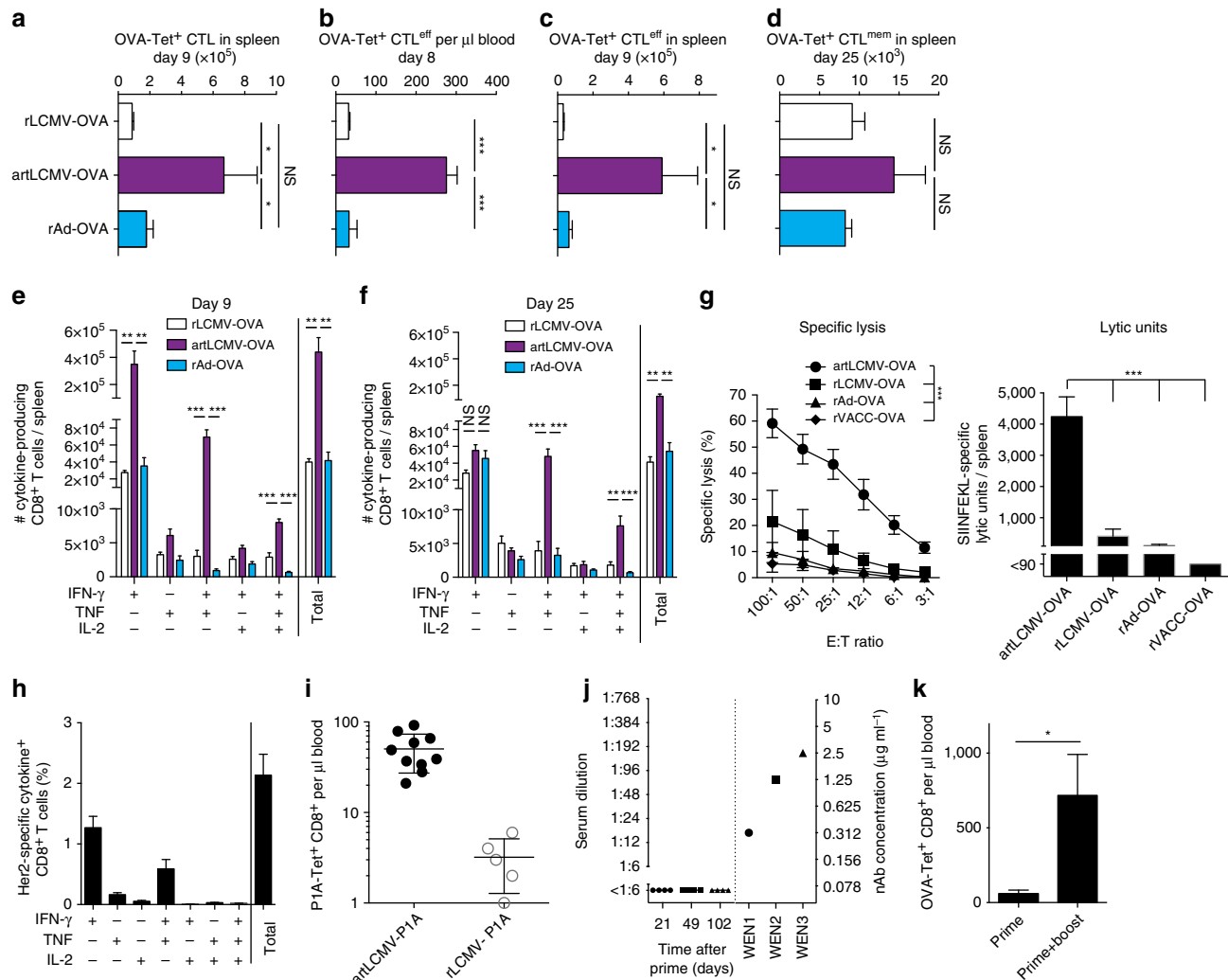

**Figure 4 | artLCMV induces polyfunctional CTL^eff responses against self and non-self antigens.** (**a–d**) OVA-specific total CTLs, CTL^eff (Klrg1⁺CD127⁻) and memory CTL (CTL^mem, Klrg1⁻CD127⁺) in blood (d8; **b**) and spleen (**a,c,d**) upon immunization with artLCMV-OVA, rLCMV-OVA or rAd-OVA. Bars represent the mean + s.e.m. (*n* = 4). *N* = 2. (**e,f**) Polyfunctional OVA-specific CTLs in spleen after artLCMV-OVA, rLCMV-OVA or rAd-OVA immunization. Bars represent the mean + s.e.m. (*n* = 4–5). *N* = 2. (**g**) OVA-specific primary *ex vivo* cytotoxicity and lytic units in spleen on d7 after artLCMV-OVA, rLCMV-OVA, rAd-OVA or rVACC-OVA vaccination. *N* = 2. (**h**) Cytokine profile of splenic Her2-specific CTLs on d9 after artLCMV-Her2 immunization. Bars represent the mean + s.e.m. (*n* = 3). *N* = 2. (**i**) P1A-specific CTLs in blood on d14 after immunization of BALB/c mice with r3LCMV-P1A or rLCMV-P1A. Symbols represent individual mice. *N* = 2. (**j**) Undetectable LCMV-neutralizing antibodies after artLCMV prime-boost immunization (prime d0, boost d87). WEN1, WEN2 and WEN3 were LCMV-neutralizing monoclonal antibody controls. Symbols show individual mice. *N* = 2. (**k**) OVA-specific CTLs in blood on d18 after prime or prime-boost immunization (d0, d14) of C57BL/6 mice with artLCMV-OVA. Bars represent the mean + s.e.m. (*n* = 3). *N* = 3. Data in **a–g** were analysed by one-way ANOVA with Bonferroni *post hoc* test and data in **k** by unpaired two-tailed Student's *t*-test. NS, not significant; *$P < 0.05$, **$P < 0.01$ and ***$P < 0.001$.

failed to prevent tumour progression, indicating that tumour antigen-specific T-cell induction was essential for clinical efficacy (Fig. 6a,b). Moreover, artLCMV-OVA-based tumour immunotherapy of EG7-OVA tumours was only effective in ST2-sufficient wt animals but failed in *Il1rl1*⁻/⁻ mice (Fig. 6c–f). These differences correlated with OVA-specific CTLs and CTL^eff, which were >tenfold higher in the blood of artLCMV-OVA-treated wt mice than in analogously treated *Il1rl1*⁻/⁻ mice (Supplementary Fig. 5c), altogether attesting to the critical function of the IL-33–ST2 axis in artLCMV-based cancer immunotherapy. To assess the potency of artLCMV-based immunotherapy in a mouse tumour model without artificially introduced non-self antigens, we exploited the P815 mastocytoma model. Immunotherapy with artLCMV expressing the cancer testis (self) antigen P1A (artLCMV-P1A) significantly delayed the growth of established

subcutaneous tumours and prolonged the animals' survival, whereas replication-deficient rLCMV-P1A and irrelevant artLCMV-GFP were ineffective (Fig. 6g,h). Of note, P815 tumours were refractory to anti-PD1 checkpoint inhibition but responded to artLCMV-P1A (Fig. 6i,j). Anti-PD1 unresponsiveness of P815 tumours has been previously documented and may be due to several mechanisms including insufficient expression of PD-1 ligands, as reported[41]. Accordingly, the combination of artLCMV-P1A and anti-PD-1 checkpoint blockade was not superior to artLCMV-P1A monotherapy (Supplementary Fig. 5h).

**artLCMV-induced CTL drive inflammatory conversion of tumours.** CTL infiltrates predict survival in many human cancers[7,9] and the recruitment of circulating specific CTLs into the tumour

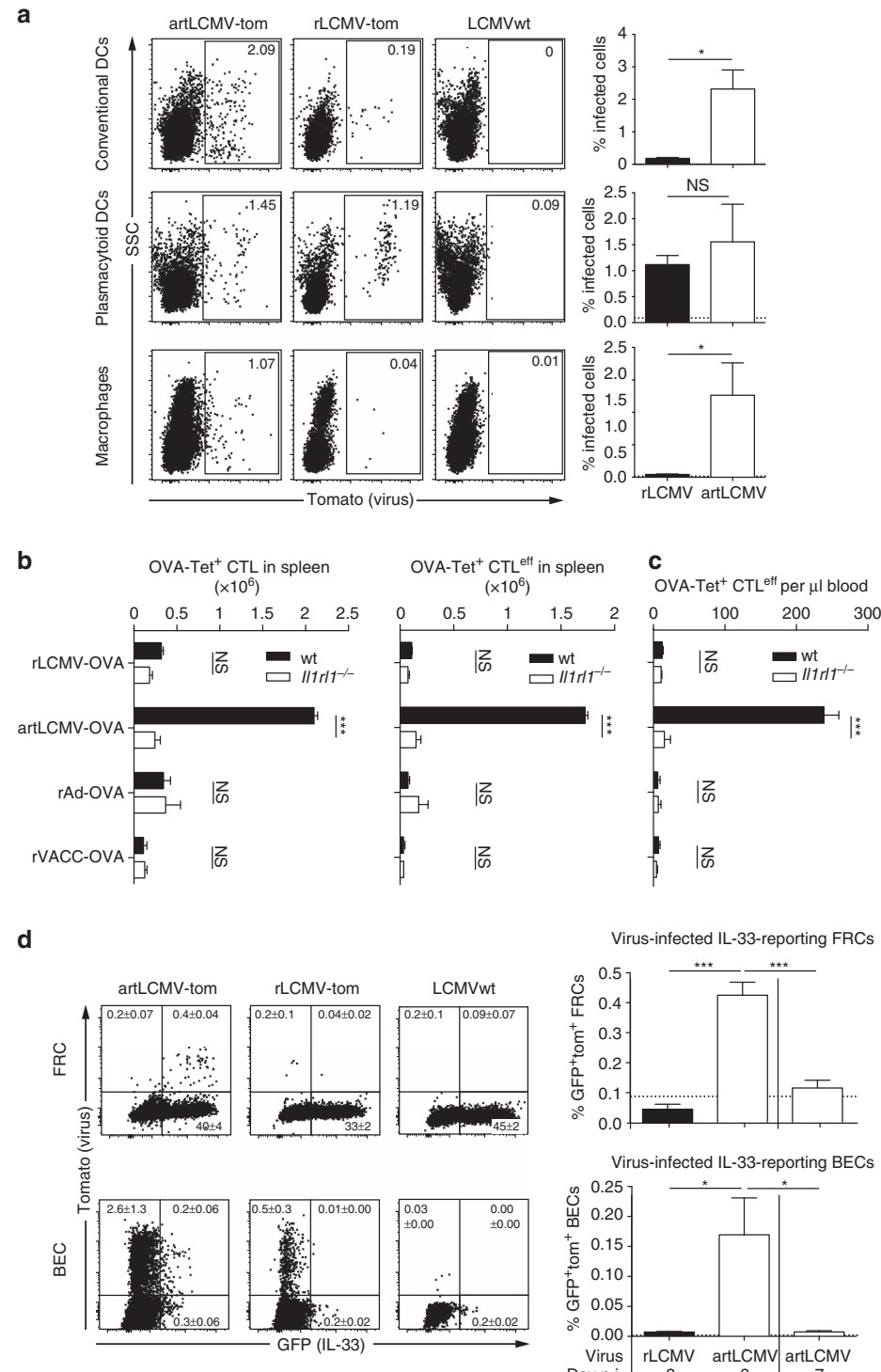

**Figure 5 | artLCMV infects DCs and IL-33-expressing stromal cells to trigger IL-33-driven CTL expansion.** (**a**) We infected C57BL/6 mice with artLCMV-tom, rLCMV-tom or LCMVwt and quantified virus-infected tomato-reporting conventional dendritic cells (lineage⁻CD11c^hi^B220⁻), plasmacytoid DCs (pDCs; lineage⁻CD11c^int^B220⁺) and macrophages (lineage⁻CD11b⁺Ly6G⁻) in spleen on d4. Representative FACS plots (left) and quantifications (right) are shown. Bars represent the mean + s.e.m. of four mice per group. $N = 2$. (**b,c**) Wt and ST2-deficient (*Il1rl1^−/−^*) mice were infected with artLCMV-OVA, rLCMV-OVA, rAd-OVA or rVACC-OVA. Spleens (**b**) were analysed on d9, peripheral blood (**c**) on d8. OVA-tetramer-binding total CD8⁺ T cells (**b**, left), as well as OVA-tetramer-binding CTL^eff^ (Klrg1⁺CD127⁻) in spleen and blood were enumerated by FACS (**b** right, **c**). Bars represent the mean + s.e.m. of four mice per group. $N = 2$. (**d**) FACS analysis of splenic fibroblastic reticular cells (FRC) and blood endothelial cells (BEC) (gated as outlined in Supplementary Figs 1 and 4c) from hemizygous IL-33 reporter mice (*IL-33^gfp/wt^*) on d3 and d7 after infection with artLCMV-tom, rLCMV-tom or LCMVwt (control). Representative FACS plots are shown. Quadrant statistics and quantifications of IL-33-expressing (GFP⁺) viral vector-infected (tom⁺) cells (bar graphs) are shown as mean + s.e.m. of four mice per group. $N = 2$. Horizontal dashed lines show technical backgrounds of uninfected controls. Data in **a**–**c** were analysed by unpaired two-tailed Student's *t*-test ((**b,c**) with Bonferroni correction) and data in **d** were analysed by one-way ANOVA with Bonferroni *post hoc* test. NS, not significant; *$P < 0.05$, ***$P < 0.001$.

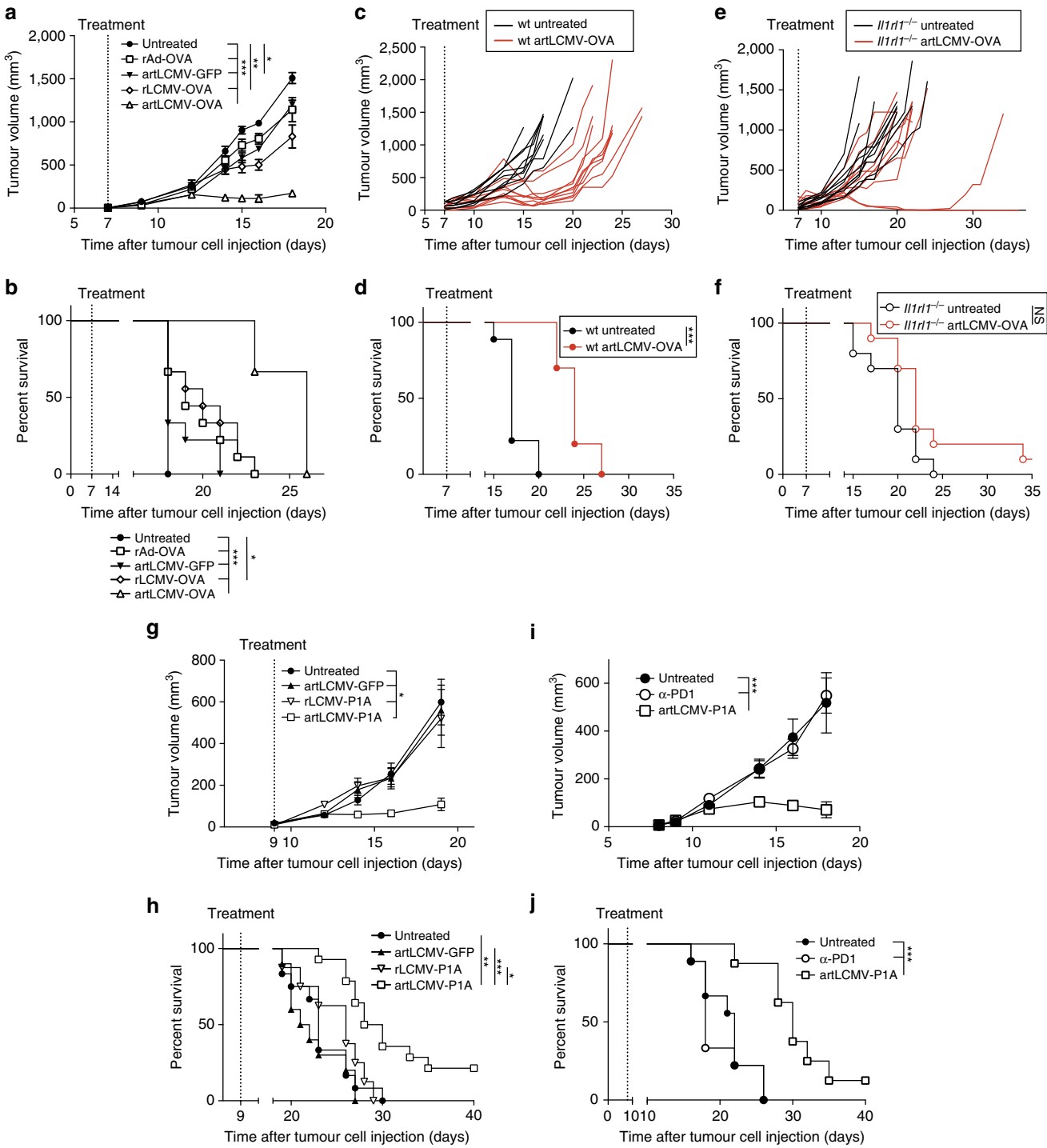

**Figure 6 | artLCMV-based immunotherapy affords antigen-specific and ST2-dependent tumour control.** (**a,b**) We implanted EG7-OVA tumour cells subcutaneously into the flank of C57BL/6 mice. On day 7, when tumours became palpable, we treated them with artLCMV-OVA, artLCMV-GFP, rLCMV-OVA, rAd-OVA or left them untreated. Tumour growth over time (**a**, terminated when the first animal was lost from follow-up owing to humane endpoint) and survival curves (**b**) are shown. Symbols represent the mean ± s.e.m. of nine mice per group. $N = 3$. (**c–f**) We implanted EG7-OVA tumour cells subcutaneously into the flank of C57BL/6 (wt, **c,d**) and ST2-deficient ($Il1rl1^{-/-}$, **e,f**) mice. When tumour masses became palpable on d7, we treated them with artLCMV-OVA (red lines) or left them untreated (black lines). Tumour growth (**c,e**, lines depict individual mice) and Kaplan–Meier survival curves (**d,f**, $n = 9$ (wt untreated), $n = 10$ (other groups)) are shown. $N = 2$. (**g,h**) We implanted P815 tumour cells subcutaneously into the flank of DBA/2 mice. When tumour masses became palpable on d9, we treated them with artLCMV-P1A, artLCMV-GFP, rLCMV-P1A or left them untreated. Tumour volumes (**g**, mean ± s.e.m.) and Kaplan–Meier survival curves based on humane endpoints (**h**) show combined data of nine (rLCMV-P1A), 12 (untreated, artLCMV-GFP) and 14 mice per group (artLCMV-P1A) from two independent experiments. $N = 2$. (**i,j**) DBA/2 mice bearing P815 tumors as in **g,h** were treated on d9 with artLCMV-P1A or 12.5 mg kg$^{-1}$ anti-PD1 antibody on d15 (earliest possible onset of the artLCMV-P1A-induced CTL response), d18, d22 and d25. Tumor volumes (**i**, mean ± s.e.m.) and survival rates (**j**) of eight (artLCMV-P1A) or nine mice per group (untreated, α-PD1) are shown. We analysed tumour growth curves in **a,g,i** by comparing the area under the curve (AUC) using one-way ANOVA with Bonferroni *post hoc* test. Survival data in **b,d,f,h,j** were analysed by log-rank tests with Bonferroni correction in **b,h,j**. NS, not significant; *$P < 0.05$, **$P < 0.01$ and ***$P < 0.001$.

represents an important goal of cancer immunotherapy. artLCMV-OVA immunotherapy yielded higher numbers of circulating OVA-specific CTLs and CTL[eff] in blood of EG7-OVA tumour-bearing mice than did artLCMV-GFP, rAd-OVA or rLCMV-OVA (Fig. 7a). Flow cytometry and immunohistochemistry documented also that artLCMV-OVA immunotherapy yielded significantly higher densities of tumour-infiltrating CTLs (TILs), OVA-specific TILs and tumour-infiltrating OVA-specific CTL[eff] (Fig. 7b,c). Analogously, artLCMV-P1A immunotherapy augmented P1A-specific CTLs and CTL[eff] in blood of tumour-bearing mice and in tumour tissue (Supplementary Fig. 5d–f). The administration of irrelevant artLCMV-GFP resulted in only modestly elevated TIL numbers and was clinically ineffective, corroborating that artLCMV immunotherapy operated in a largely antigen-specific manner (Fig. 7b,c). Accordingly, immunohistochemical analysis did not reveal artLCMV antigen in the tumour, while viral antigen was

clearly detected in spleen (Supplementary Fig. 5g), arguing against an oncolytic effect of artLCMV. Inflammatory gene expression profiling of EG7-OVA tumours on day 9 after artLCMV-OVA immunotherapy identified 30 genes, which were significantly different from tumours of untreated control mice. Thereof 20 were IFN-inducible (Fig. 7d). Validation by TaqMan RT-PCR confirmed that artLCMV-OVA immunotherapy induced several inflammatory mediators namely including the chemokines Ccl5, Cxcl9, Ccl4 and Cxcl10, which are predictive of prolonged survival in human cancers (Fig. 7e)[42–44]. Of note, expression of CCL5 was significantly higher in tumours of artLCMV-OVA-treated animals than in those receiving rLCMV-OVA or rAd-OVA (Fig. 7e, similar trend for Cxcl9). Importantly, however, only tumours of artLCMV-OVA immunized but not of artLCMV-GFP control-treated animals exhibited substantial chemokine induction. This indicated that OVA-specific CTL[eff] infiltration was essential for

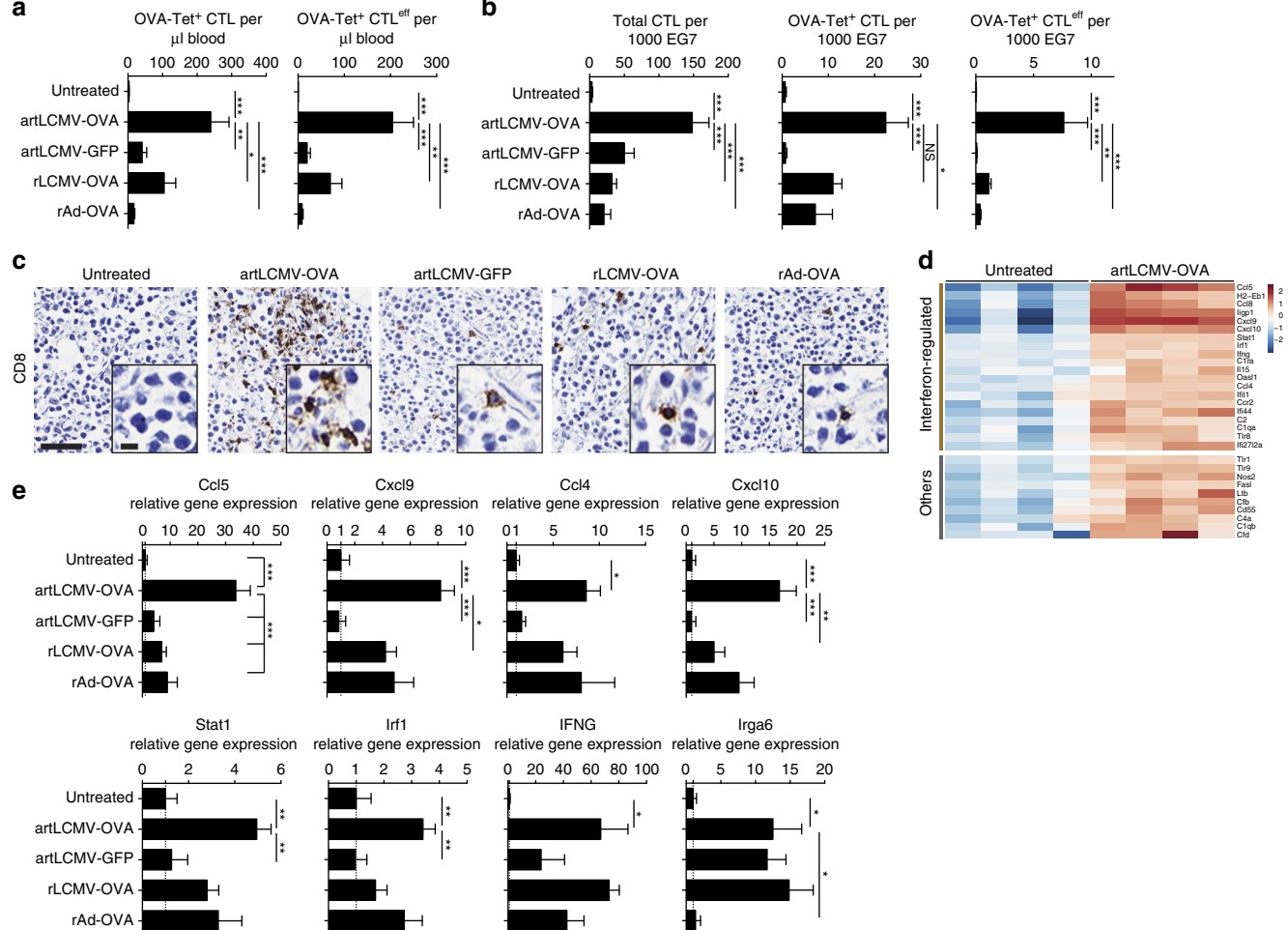

**Figure 7 | artLCMV immunotherapy leads to CTL infiltration and inflammatory conversion of the tumour. (a–e)** We implanted EG7-OVA tumour cells subcutaneously into the flank of C57BL/6 mice. On day 7, when tumours became palpable, we treated them with artLCMV-OVA, artLCMV-GFP, rLCMV-OVA, rAd-OVA or left them untreated. Peripheral blood (**a**) and tumour tissue (**b–e**) were analysed on d8 and d9 after treatment, respectively. We enumerated OVA-tetramer-binding total CTLs and CTL[eff] (Klrg1[+]) in blood (**a**) and tumour (**b**) as indicated. CTLs were gated as live CD8[+]CD4[−]CD3[+]B220[−] lymphocytes. For normalization to EG7 tumour cells the latter were differentiated from infiltrating inflammatory cells by size and granularity. CTL[eff] were identified as Klrg1[+]. Bars represent the mean + s.e.m. of five (artLCMV-OVA) or four mice (all other groups). $N = 2$.
(**c**) Tumours from mice as in **a,b** were analysed for infiltrating CD8[+] cells by immunohistochemistry. Representative pictures from four to five mice as in **b** are shown. Scale bars 10 μm (inset) and 100 μm (overview). $N = 2$. (**d**) Gene expression profiles of inflammation-associated genes from tumour tissue of untreated or artLCMV-OVA-treated animals as in **a–c** by Nanostring technology. Differentially expressed genes (fold change $\geq 2$, adjusted $P$ value $< 0.05$) are displayed. Each lane represents a tumour from an individual mouse. Interferon-regulated genes are indicated according to www.interferome.org.
(**e**) TaqMan RT-PCR validation of select genes identified in **d**. Bars represent the mean + s.e.m. of 4–5 mice as in **b**. Data in **a,b** and **e** were analysed by one-way ANOVA with Dunnett's *post hoc* test. NS, not significant; *$P < 0.05$, **$P < 0.01$ and ***$P < 0.001$.

inflammatory activation of the tumour microenvironment in artLCMV-OVA immunotherapy whereas the accompanying non-specific inflammation, which was also induced by artLCMV-GFP, was insufficient to mediate these effects in full.

## Discussion

These results identify virally delivered alarmin signals as key drivers of protective CTL[eff] responses in vectored cancer immunotherapy. The function of such damage-associated molecular patterns appears non-redundant with PAMPs, which can also be provided by replication-deficient viral delivery systems[45,46]. As schematically outlined in Supplementary Fig. 6, the ability of artLCMV to trigger IL-33/ST2-dependent CTL[eff] induction correlated with the vectors' spread into IL-33-expressing lymphoid stroma cells. Subsequent death of these cells, possibly as a consequence of CTL attack[47], offers a likely mechanism for IL-33 release to neighbouring T cells. Activated CTLs inside secondary lymphoid organs express ST2, and IL-33 sensing is known to potentiate their expansion, effector differentiation and survival[11]. The present observations suggest that upon emigration to the tumour, these alarmin-imprinted CTLs contribute essentially to tumour microenvironment changes and to tumour control, thus providing a mechanistic rationale for the exploitation of replicating delivery platforms in the fight against cancer. Besides renewed interest in the tumour immunotherapy field, replicating live vaccine approaches currently experience a revival in indications such as Ebola hemorrhagic fever[48], tuberculosis[49] and immunodeficiency virus infection[50]. These infectious diseases and cancer have in common that potency is rate-limiting, outweighing potential concerns related to the safety profile of replicating vector systems or to the release of genetically modified organisms into the environment. Accordingly, the safety profile of genetically engineered live viruses has become acceptable for oncolytic virus therapy, with a licensed product already on the market[1].

The development of artLCMV represents an innovative addition to a limited quiver of replicating viral vaccine delivery systems. Several mechanistic features suggest it holds promise for tumour immunotherapy. Efficient DC targeting results in potent CTL priming. In vivo spread and infection of IL-33-expressing lymphoid stromal cells unleashes the IL-33–ST2 alarmin pathway, thereby augmenting CTL function and effector differentiation[11]. Low LCMV seroprevalence in the human population[21,22] predicts high response rates. Inefficient vector-neutralizing antibody induction[38] facilitates repeated vector re-administration. The intravenous administration of LCMV is safe in humans and non-human primates[18,51].

In recent years, checkpoint inhibitors such as anti-PD-1/PD-L1 or anti-CTLA-4 antibodies had groundbreaking success in a variety of malignancies[2]. However, immune checkpoint blockade merely disinhibits ongoing T-cell responses and, by consequence, tends to fail in tumours with a paucity of pre-existing tumour-infiltrating CTLs[52,53]. Such 'cold' tumours can be the result of immunoediting and T-cell escape[54]. Alternatively, tumours can exhibit immunogenic determinants but fail to induce clinically significant CTL responses, which is referred to as immune exclusion or ignorance[14,53]. As a powerful tool for active immunization artLCMV immunotherapy delivered significant numbers of tumour-reactive CTL[eff] to the tumour bed, resulting in an inflammatory conversion of the tumour microenvironment (Fig. 7, Supplementary Fig. 5). This transition is thus predicted to render the corresponding tumours more responsive to immunomodulatory therapy.

Beyond the commonly benign nature of human LCMV infection and a lack of horizontal transmission in humans, inefficient co-packaging of the three artLCMV genome segments represents a molecularly defined mechanism of attenuation. The >1,000-fold increase in i.c. mouse $LD_{50}$ exceeds the ~100-fold safety margin of the clinically used, live-attenuated Junin arenavirus vaccine Candid#1 (ref. 55). Last but not least, genetic stability (Fig. 2a–d) should enable industrial exploitation. During scale-up in batch production, but also upon administration to vaccinees, rearranged genomes with a fitness gain are readily selected[32,56], compromising product safety as well as efficacy[29,30]. Accordingly, stable transgene expression and attenuation of artLCMV, together with the simplicity and rapidity of vector generation, represent critical assets for clinical translation and the vector's exploitation in personalized medicine approaches[57].

Taken together, our study identifies alarmin signals as crucial for the induction of protective anti-tumour CTL[eff] responses in vectored immunotherapy. By demonstrating mechanistically that alarmin release depends on viral in vivo spread, our work suggests that live viruses deserve their place in the rapidly evolving array of cancer treatment modalities. artLCMV may represent the prototype of a novel class of live microbial delivery systems, which leverage not only PAMPs but also alarmin release for CTL[eff] differentiation and enhanced anti-tumour efficacy.

## Methods

**Cells.** BHK-21 cells were obtained from ECACC (Clone 13, Cat #85011433), MC57 cells (CRL-2295), EL-4 cells (TIB-39), EG7 thymoma cells (EL-4 cells expressing OVA, CRL-2113) and P815 mastocytoma cells (TIB-64) were obtained from ATCC. Stably transfected BHK-21 cells expressing the LCMV-NP and -GP proteins, respectively (BHK-NP; BHK-GP), and GP-expressing 293T cells have previously been described[26,58]. All cell lines were tested mycoplasma negative.

**Viruses and virus neutralization test.** The origin, passage and titration of the LCMV strains Armstrong and Clone 13 strains have been described[11]. GP-IFF has previously been described[58]. For GFP-IFF we used rat-anti-GFP antibody (Biolegend, 1:2,000 dilution) as primary antibody. Replication-competent LCM viruses were propagated on BHK-21 cells, rLCMV vectors on 293T-GP cells[26]. Growth curves of LCM viruses were performed on BHK-21 cells at a multiplicity of infection of 0.01. The reverse genetic engineering of rLCMV, r3LCMV and artLCMV vectors has been described[26,27]. Entire ORFs of the cancer-testis antigen P1A (comprising the immunodominant LPYLGWLVF epitope), GFP, dTomato, OVA (comprising the immunodominant SIINFEKL epitope) and Cre were used for insertion into the respective vectors and viruses. artLCMV-Her2 contained a ubiquitin-based expression cassette[59] delivering the immunodominant TYLPANASL miniepitope. Recombinant vaccinia virus and E1-deleted recombinant adenovirus 5-based vectors expressing ovalbumin (rVACC-OVA, rAd-OVA) have previously been described[60,61]. A genetically tagged GPC-encoding S segment cDNA (pol-I-S-GPC[tag]) was generated by codon-optimizing the 255 C-terminal nucleotides of the viral GPC ORF, creating a novel ORF named 'GPC[tag]'. To create a $S_{rec}$#1 expression plasmid, we substituted the IGR in pol-I-S-GPC[tag] for a synthetic cDNA fragment (Genscript) as described in Supplementary Fig. 2e. The neutralizing capacity of mAb and immune serum was analysed in immunofocus reduction assays[11].

**Animal experimentation and ethics statement.** Il33[−/−] mice[62] were obtained through the RIKEN Center for Developmental Biology (Acc. No. CDB0631K; http://www.cdb.riken.jp/arg/mutant%20mice%20list.html). AGRAG mice (IFNα/βR[−/−], IFNγR[−/−], RAG1[−/−] triple-deficient)[63], Il1rl1[−/−] (ref. 64) and ZP3-Cre mice[65] have been described. IL33 reporter mice (Il33[gfp/wt]) were generated by intercrossing IL-33[−/−] mice with the ZP3-Cre germ line deleter strain to remove the neomycin cassette. C57BL/6, BALB/c and DBA/2 wild-type mice were either purchased from Charles River and Janvier Labs or were bred at the Institut für Labortierkunde of the University of Zurich, Switzerland under specific pathogen-free conditions. Unless specified otherwise, rAd-OVA vectors and rVACC-OVA were administered in their respective optimal dose range of 10[8] viral particles i.v. and 10[6] PFU i.v., respectively[66,67]. The same respective doses were used for intramuscular (rAd-OVA) and intradermal or subcutaneous (rVACC-OVA) immunizations in the experiments to Supplementary Fig. 2. artLCMV and r3LCMV were routinely given at a dose of 10[5] PFU i.v., except for virus tracing experiments in C57BL/6 mice (10[6] PFU) and experiments in AGRAG mice (10[4] PFU). rLCMV vectors were used at 10[5] PFU for standard immunization experiments or at 10[6] PFU for immunization with tumour self antigens, for tumour immunotherapy and for Cr[51] release experiments. Except for prime-boost immunization experiments in Fig. 4j,k, animals were given only a single shot of viral vector or virus, respectively. LCMVwt was administered i.v. at the same dose

as artLCMV was used. Intracranial LCMV challenge was performed through the skull and animals developing terminal disease were euthanized by $CO_2$ inhalation in accordance with the Swiss law. PD-1 blocking antibody (clone RMP1-14, from BioXcell) was administered at a dose of 12.5 mg kg$^{-1}$ intraperitoneally. CD8+ T cells were depleted by injecting 200 µg anti-CD8 antibody (YTS169, from BioXcell) intraperitoneally. Animal experiments were performed at the Universities of Geneva and Basel in accordance with the Swiss law for animal protection. Permission was granted by the Direction générale de la santé, Domaine de l'expérimentation animale of the Canton of Geneva and by the Veterinäramt Basel-Stadt, respectively. Experimental groups were sex- and age-matched. Animals in tumour therapy experiments were assigned to groups in a manner to assure even distribution of tumour volumes between groups at the time of tumour therapy. The groups were neither randomized, nor were experiments conducted in a blinded manner. The Swiss law for animal protection requires that mice with wounds on the tumour or exhibiting signs of distress (evident namely in lethargy, hunchback, piloerection, emaciation and agonal breathing) be euthanized by $CO_2$ inhalation irrespective of tumour size and diameter. Accordingly, animals not reaching humane endpoints of tumour volume or diameter were excluded from survival curves.

**Virus sequencing and gene expression profiling.** Viral RNA was extracted from cell culture supernatant or serum of infected mice using the QIAamp Viral RNA Mini Kit (QIAGEN, cat. no. 52906). Reverse-transcription was performed with ThermoScript RT-PCR System (Invitrogen) and an LCMV-*NP*-specific primer (5′-GGCTCCCAGATCTGAAAACTGTT-3′). PCR amplification used the same primer together with a *GPC*-specific primer (5′-GCTGGCTTGTCACTAATGG CTC-3′). Amplified products were purified for Sanger sequencing (Microsynth). Whole-cell RNA was extracted from tumour tissue using QIAzol (QIAGEN). nCounter Nanostring Mouse Inflammation v2 assay and Applied Biosystems TaqMan RT-PCR assays were used for quantification of gene expression. TaqMan results were normalized to GAPDH.

**Flow cytometry.** Antibodies against CD4 (RM4-5 or GK1.5), CD8 (53-6.7), CD45R/B220 (RA3-6B2), CD45.2 (104), Ter-119 (TER-119), CD31 (390), gp38 (Podoplanin; 8.1.1), CD3 (17A2), Klrg1 (2F1), CD127 (A7R34), CD11c (N418), CD11b (M1/70), CD19 (6D5), NK1.1 (PK136), CD90.2 (30-H12), GR-1 (RB6-8C5), IFN-γ (XMG1.2), TNF (MP6-XT22) and IL-2 (JES6-5H4) were from Biolegend, Pharmingen and eBioscience. All fluorescently labelled monoclonal antibodies were used at a 1:100 dilution, except for gp38 (Podoplanin) and Ter-119, which were diluted 1:1,000, and 1:10, respectively. Dead cells were excluded with Zombie UV Fixable Viability Kit (Biolegend, cat. no. 423108). OVA- (SIINFEKL) and P1A epitope- (LPYLGWLVF) specific CTLs were identified as peptide-MHC class I tetramers (TCMetrix) binding cells amongst CD8+B220− lymphocytes. BD Trucount Absolute Counting Tubes were used to determine absolute cell counts in blood. Spleen lymphocyte counts were determined in a Neubauer chamber. Peptide-MHC tetramer-binding CTLs were back calculated. Cytokine profiles after restimulation with peptide (SIINFEKL for OVA, TYLPANASL for Her2; ProImmune) were determined in intracellular cytokine assays[11]. Samples were measured on BD LSRFortessa and Beckman Coulter Gallios flow cytometers and were analysed using FlowJo software (Tree Star, Ashland, OR).

**Isolation of cells.** Splenic single-cell suspensions were prepared by mechanical disruption. The isolation of stromal cells from spleen has been described[68]. Briefly, spleens were flushed with RPMI containing 3 mg ml$^{-1}$ Collagenase IV (Worthington), 40 µg ml$^{-1}$ DNAseI (Roche) and 2% (vol/vol) FCS, were cut into pieces and digested. Erythrocytes were lysed in 0.15 M NH$_4$Cl/10 mM KHCO$_3$/0.1 mM EDTA and haematopoietic CD45+ cells were depleted using anti-CD45 beads (Miltenyi Biotec). For TIL analysis, tumours were cut and digested with accutase (PAA), Collagenase IV (Worthington), Hyaluronidase (Sigma) and DNAseI (Roche) and red blood cells were lysed. Mononuclear cells were isolated using a Histopaque-1119 gradient.

**Tumour transplantation and measurements.** Suspensions containing $10^6$ tumour cells were implanted subcutaneously in the right flank. Tumour growth was monitored three times per week and the longest ('length') and the shortest diameter ('width') were measured using a caliper. Tumour volumes (mm$^3$) were calculated as 1/2 (length*width^2). In accordance with the Swiss law for animal protection, mice were euthanized when tumour volumes exceeded 1,500 mm$^3$ or when the longest median tumour diameter exceeded 20 mm.

**Immunohistochemistry.** For immunohistochemical bright-field staining, tissues were prepared in HOPE fixative (DCS Innovative) and paraffin embedded. Upon inactivation of endogenous peroxidases, tissue sections were incubated with rat-anti-mouse CD8a (YTS169.4, diluted 1:2,000) or rat-anti-LCMV NP (VL-4, hybridoma supernatant was diluted 1:10). Bound primary antibodies were visualized with biotin-labelled anti-rat antibody (diluted 1:100) and strepta-vidin-peroxidase staining method using polymerized 3,3′-diaminobenzidine (all reagents from Dako; haemalaun counterstaining of nuclei).

For detection of GFP in virus-infected organs, cryosections (10 µm) were prepared from PFA-fixed tissues and nuclei were stained with 4′,6-diamidino-2-phenylindole (DAPI, Invitrogen).

Immunostained bright-field sections and fluorescence sections of virus-infected tissues were scanned using a Panoramic Digital Slide Scanner 250 FLASH II (3DHISTECH). White balance was adjusted and contrast was linearly enhanced using the tools 'levels', 'curves', 'brightness' and 'contrast' in Photoshop CS6 (Adobe).

For GFP detection in naïve *IL-33*$^{gfp/gfp}$ and *IL-33*$^{gfp/wt}$, cryosections (8–10 µm) were prepared from PFA-fixed spleens. For detection of nuclear IL-33, rat anti-gp38 (anti-podoplanin, 8F11, MBL, diluted 1:900), rabbit anti-GFP (A11122, Invitrogen, diluted 1:400) and goat anti-IL33 (AF3626, R&D Systems, diluted 1:200) were used as primary antibodies. As secondary antibodies donkey-anti-rat IgG HRP (horse radish peroxidase, diluted 1:500), donkey-anti-rabbit Alexa488 (diluted 1:500) and donkey-anti-goat Alexa647 (diluted 1:300) were used (Jackson Immunoresearch). HRP was visualized using a Tyramide Cy3 (Invitrogen) reaction. Nuclei were stained with DAPI. Immunostainings were visualized using an AxioImager Z.1 microscope (Zeiss), followed by image processing in Adobe Photoshop.

**Primary *ex vivo* cytotoxicity assays.** Primary *ex vivo* cytotoxicity of splenocytes was determined in a 5-h assay on Cr51-loaded EL-4 target cells pulsed with SIINFEKL peptide according to established procedures[69]. Cytotoxicity on unlabelled target cells was subtracted to obtain specific lysis. Lytic units were determined according to established procedures from dose–response curves[69]; one lytic unit was calculated to represent the minimal number of spleen cells that lysed >10% of the target cells.

**IFN-α ELISA.** Serum interferon-alpha levels were determined by ELISA using the Verikine Mouse Interferon Alpha ELISA Kit (PBL Assay Science, cat. no. KMC4011).

**Statistical analysis.** For statistical analysis, GraphPad Prism software (Version 6.0, GraphPad Software) was used. Differences between two groups were assessed using unpaired two-tailed Student's *t*-tests. Single values of multiple groups were compared by one-way analysis of variance (ANOVA), followed by Bonferroni *post hoc* test or Dunnett's *post hoc* test when comparing against a reference group. A Bonferroni correction was made when comparing multiple parameters. Two-way ANOVA with Bonferroni *post hoc* test was used to compare multiple groups with multiple measurements. Survival curves were analysed by log-rank tests with Bonferroni correction. For the comparison of tumour growth curves, the area under the curve was compared[70]. The analysis ended when the first animals reached human endpoints.

Viral load data were log-converted to obtain a near-normal distribution prior to statistical analysis. Variances within different groups in a given experiment were similar. In accordance with current standard practice in the field of viral immunology, only very substantial differences were reported, obviating the need for variance testing.

*P* values of $P < 0.05$ were considered significant (*), $P < 0.01$ (**) and $P < 0.001$ (***) as highly significant.

**Data availability.** Nanostring data that support the findings of this study have been deposited in National Center for Biotechnology Information Gene Expression Omnibus (GEO) with the accession code GSE84039. The nucleotide sequences of recombinant and recombined LCMV segments have been deposited in GenBank with the accession codes KX462116-KX462128. All relevant data are available from the authors upon request.

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

## Acknowledgements

We wish to thank Patricia Aparicio-Domingo, Kerstin Narr, Claire-Anne Siegrist and Paul-Henri Lambert for helpful discussions, Nadège Lagarde, Severine Clement and Karsten Stauffer for excellent technical assistance, A. McKenzie for *Il1rl1*<sup>−/−</sup> mice (obtained under MTA), M. Groettrup (University of Constance) for VACC-OVA, originally generated by J. Yewdell (National Institute of Allergy and Infectious Diseases) and D. von Laer (University of Innsbruck) for GP-expressing 293T cells. Furthermore, we wish to thank Didier Chollet of the iGE3 genomics platform of the University of Geneva, Florian Geier of the Bioinformatics Core Facility (of the Department of Biomedicine, University of Basel) and the Center for Scientific Computing (SciCore, University of Basel) for their support with gene expression profiling and biomathematical analysis. This work was supported by the Klaus Tschira Stiftung gGmbH (to D.D.P. and D.M.), by the Swiss National Science Foundation (Sinergia Grant No. 310030_149340 to D.D.P., D.M., S.A.L. and M.L.) and by Hookipa Biotech AG (to D.D.P.). D.M. holds a stipendiary professorship of the Swiss National Science Foundation (No. PP00P3_152928).

## Author contributions

S.M.K., S.D., W.V.B., M. Kreutzfeldt, N.P., P.M., M. Kreuzaler, F.K., M.L., S.A.L., A.Z., D.M. and D.D.P. conceived and designed the experiments. S.M.K., S.D., W.V.B., M. Kreutzfeldt, N.P., P.M., M. Kreuzaler, M.L. and S.F. performed the experiments. S.M.K., S.D., W.V.B., M. Kreutzfeldt, P.M., A.Z., D.M. and D.D.P. analysed the data. S.M.K., A.Z., D.M. and D.D.P. wrote the paper.

## Additional information

**Competing interests:** D.D.P. is a shareholder and also a consultant to Hookipa Biotech AG commercializing arenavirus-based vector technology. S.M.K., S.D., MarK, N.P., D.M. and D.D.P. are listed as inventors on a patent entitled 'Tri-segmented arenaviruses as vaccine vectors' (application number PCT/EP2015/076458) describing artLCMV-based vector technology. The remaining authors declare no competing financial interests.

