## [Peer Review File · Nature Communications]

Reviewers' comments:

Reviewer #1 (Remarks to the Author):

This manuscript describes a novel LCMV recombinant vector for transgene expression and therapeutic development. In the report, the authors nicely demonstrate how the virus can encode tumor-associated antigens for use as a cancer vaccine. The recombinant virus is replication competent yet attenuated and the paper demonstrates that the vector can induce cytotoxic T cell responses, promote anti-tumor immunity through infection of conventional dendritic cells in an IL33-ST2-dependent manner and induce limited neutralizing antibody titers against the vector. The recent focus on viral vectors for both oncolytic therapy and vaccination makes this report timely and interesting. The manuscript is generally well written and contains several innovative findings, including a method for improving LCMV as a therapeutic agent and the finding that IL-33 is involved in the anti-tumor effects. The data is well presented and has implications for promoting influx of tumor-reactive T cells to native "cold" tumors lacking T cell infiltration, an important goal in current tumor immunotherapy. I do, however, have some minor issues for the authors to consider as detailed below.

Specific Comments

1. The characterization of the vector is well described and the immunology data is very intriguing with potent CTL responses and minimal neutralizing antibody induction. It was a little unclear from the figures, however, on the frequency of virus administration. Could the authors confirm that in the murine experiments only one vaccination was given, except for the prime-boost experiment?
2. Although the authors use ovalbumin as an experimental system, this is likely less appropriate than using a more physiologically relevant tumor antigen. The use of HER-2 and P1A do add support to the underlying hypotheses. Figure 4j shows minimal increased survival with the LCMV-P1A vector and that anti-PD-1 does not work. Did they consider combining the tumor antigen vector and anti-PD-1 in combination? Further, would repeated virus administration improve the therapeutic activity?
3. The authors provide a novel mechanism of anti-tumor activity, in which they propose that the LCMV vector infects dendritic cells and IL-33-secreting fibroblast reticular cells (suppl. Figure 2c). This is interesting and the authors go on to conclude that IL-33 is altering the tumor microenvironment chemokine milieu (Fig. 5). How does this happen? I wonder if you treated the mice with exogenous IL-33 would a similar change in the tumor microenvironment occur? These findings, which appear highly specific for the LCMV vector, are a bit unexpected since IL-33 has been shown to preferentially induce Th2-type cytokines. I might also suggest looking for Th1 vs. Th2 cytokines in virus-treated animals to better understand how IL-33 is mediating the modifications of the tumor microenvironment.
4. In Figure 2a there appear to be two lines for the LCMVwt titers. Is this a mistake?
5. In figure 2d the authors show that interferon release is greatest for the artLCMV vector. While this is interesting, it was very surprising that there was no interferon release with the vaccinia virus vector and very low for the rLCMV vector. Were all these viruses given by IV administration? It may be worth showing the vaccinia virus induces interferon by alternate routes of administration, such as subcutaneous or tail vein scratch. These results suggest that the vaccinia is attenuated in some way and perhaps would not be a good control in the other experimental settings in which it has been used.

Reviewer #2 (Remarks to the Author):

In this study the group of Kallert et al report on the development and characterization of a new attenuated, replicating vaccine vector based on molecular modification of the ambisense RNA virus

LCMV.

The authors start by demonstrating the make-up of the vector as well as the attenuation and genetic and phenotypic stability of their vector platform. Next the immunogenicity of the new vector platform is compared to that of a few other vaccine platforms using a several different types of analyses. As a mechanistic reason for a better induction of CTL responses, the authors demonstrate that the new vector – unlike the original non-replicating variant – induces IL-33, and that a much reduced T-cell response is found in ST2 deficient mice, lacking the relevant receptor for this cytokine. Then the vector is studied in several relevant tumor model systems, and the superiority of the new vector is clearly established. Finally, they show that the primed antigen-specific CD8 T cells infiltrate the tumor site and create an inflammatory site.

Overall, this is an interesting paper on a novel type of vaccine vector, well-written and based on a very clear and logic strategy. The results are convincing and highly pertinent. From what is presented it is clear that the new vector platform can be applied to induce strong CD8 T-cell responses, albeit I do not believe the authors have been absolutely fair in their selection of vector systems to be used for comparisons, e.g. i.v. injection of adenovectors is not the ideal route and, similarly, the vaccinia vector should perhaps have been given intradermally for a more relevant comparison.

The authors claim that vector immunity do not substantially inhibit booster response, and I tend to believe them. However, in order to really prove their point demonstrating absence of vector specific antibodies do not suffice. The real test is to vaccinate a group of mice say twice with a vector directed towards X, and then come back and vaccinate such mice with a vector directed towards Y; by comparing the CTL response against the Y antigen in this group to that in a group of previously unvaccinated mice, you will get a clear picture of how much the vector immunity can interfere with subsequent immunization. Arguably, the capacity to inhibit a secondary/tertiary response might differ from that of reducing a primary response, but this seems to the best way possible, unless an adoptive is taken.

Finally, in the abstract it very indirectly implied that the effector cells raised by the new vector is somehow unique working in part by turning a cold tumor in a hot one. Personally, I tend to believe that this is more a matter of numbers of CTLs available and attracted, not a difference in quality. Unless, the authors want to make this an issue and do comparative adoptive transfer s of matched numbers of effector cell , I suggest the sentence is modified to include the possibility that this phenomenon could be a quantitative issue as much as a qualitative one.

REVIEWERS' COMMENTS:

Reviewer #1 (Remarks to the Author):

The authors have responded well to the previous critiques, and now include a new experiment (Suppl. Figure 2) demonstrating the absence of IFN with vaccinia virus administered through alternative routes of delivery as suggested by both reviewers. They have also amended several statements in the paper better reflecting the data presented. I think this paper will be of significant interest to many readers.

Reviewer #2 (Remarks to the Author):

The authors have addressed my comments in a satisfactory way, and I have no additional queries

** See Nature Research's author and referees' website at www.nature.com/authors for information about policies, services and author benefits

Point-by-point reply to the reviewers' comments

Reviewer #1 (Remarks to the Author):

This manuscript describes a novel LCMV recombinant vector for transgene expression and therapeutic development. In the report, the authors nicely demonstrate how the virus can encode tumor-associated antigens for use as a cancer vaccine. The recombinant virus is replication competent yet attenuated and the paper demonstrates that the vector can induce cytotoxic T cell responses, promote anti-tumor immunity through infection of conventional dendritic cells in an IL33-ST2-dependent manner and induce limited neutralizing antibody titers against the vector. The recent focus on viral vectors for both oncolytic therapy and vaccination makes this report timely and interesting. The manuscript is generally well written and contains several innovative findings, including a method for improving LCMV as a therapeutic agent and the finding that IL-33 is involved in the anti-tumor effects. The data is well presented and has implications for promoting influx of tumor-reactive T cells to native "cold" tumors lacking T cell infiltration, an important goal in current tumor immunotherapy. I do, however, have some minor issues for the authors to consider as detailed below.

Specific Comments

1. The characterization of the vector is well described and the immunology data is very intriguing with potent CTL responses and minimal neutralizing antibody induction. It was a little unclear from the figures, however, on the frequency of virus administration. Could the authors confirm that in the murine experiments only one vaccination was given, except for the prime-boost experiment?

The revised manuscript's Methods section specifies that "except for prime-boost immunization experiments in Fig. 1n-o, animals were given only a single shot of viral vector or virus, respectively" (lines 376-378).

2. Although the authors use ovalbumin as an experimental system, this is likely less appropriate than using a more physiologically relevant tumor antigen. The use of HER-2 and P1A do add support to the underlying hypotheses. Figure 4j shows minimal increased survival with the LCMV-P1A vector and that anti-PD-1 does not work. Did they consider combining the tumor antigen vector and anti-PD-1 in combination?

In response to the reviewer's question we have included Supplementary Figure 4h, showing that artLCMV-P1A plus anti-PD-1 combination therapy was not superior to artLCMV-P1A monotherapy in treating P815 tumors. In the respective text section (lines 248-251) we explain that anti-PD1 unresponsiveness of P815 tumors has been reported to result from insufficient expression of PD-1 ligands (Iwai et al. Proc Natl Acad Sci U S A. 2002;99(19):12293-7). Additional mechanisms may comprise alternative immune

checkpoints but also cellular immunomodulation, and in combination may account for the lack of synergy of anti-PD-1 with artLCMV in this model.

Further, would repeated virus administration improve the therapeutic activity?

According to our preliminary data, a minimal interval of 14 days between artLCMV prime and boost must be respected for a clear booster effect to occur. Moreover, the booster response takes a few days to set in (see Fig. 2o, booster response measured on day 18 after prime, which is day 4 after boost). Regrettably, most artLCMV-treated mice reached human endpoints within about this time frame, which precluded us from conclusively testing the therapeutic efficacy of a booster vaccination in the tumor models used in our study.

3. The authors provide a novel mechanism of anti-tumor activity, in which they propose that the LCMV vector infects dendritic cells and IL-33-secreting fibroblast reticular cells (suppl. Figure 2c [revised Supplementary Fig. 3c]). This is interesting and the authors go on to conclude that IL-33 is altering the tumor microenvironment chemokine milieu (Fig. 5). How does this happen? I wonder if you treated the mice with exogenous IL-33 would a similar change in the tumor microenvironment occur? These findings, which appear highly specific for the LCMV vector, are a bit unexpected since IL-33 has been shown to preferentially induce Th2-type cytokines. I might also suggest looking for Th1 vs. Th2 cytokines in virus-treated animals to better understand how IL-33 is mediating the modifications of the tumor microenvironment.

- As evident from the “irrelevant antigen” artLCMV-GFP control group, the non-specific inflammatory reaction (including IL-33), which was triggered by artLCMV vectors, was insufficient to drive significant tumor microenvironment changes (Fig. 5e). Substantial chemokine induction, for example, occurred only when tumor antigen-specific T cell responses were induced (artLCMV-OVA group in Fig. 5e). This argued against direct IL-33 effects on the tumor microenvironment as a main underlying mechanism. Accordingly, and in agreement with published literature (Bonilla et al. Science. 2012;335(6071):984-9), we consider it most plausible that IL-33 is acting at the level of tumor antigen-specific T cell expansion / differentiation inside secondary lymphoid organs (Fig. 3b), where artLCMV-infected IL-33-expressing fibroblastic reticular cells and endothelial cells were identified (Fig. 3d). We have reworked the manuscript for more clarity on these points (lines 211, 228, 273-278, 283-292), and have included a schematic summary of the proposed chain of events in the new Supplementary Fig. 5.

- We have revised the introduction section to better emphasize that several independent high-ranking publications from our lab and others have clearly established IL-33 as a key driver of type 1 immune response in the context of viral infections and GVHD (Bonilla et al. Science. 2012;335(6071):984-9, Baumann et al. Proc Natl Acad Sci U S A. 2015;112(13):4056-61, Reichenbach et al. Blood. 2015;125(20):3183-92, Rood et al.

Blood. 2016;127(4):426-35), Sesti-Costa J Immunol. 2013;191(1):283-92, recently reviewed in Peine et al. Trends Immunol. 2016;37(5):321-33) (lines 70-75).

4. In Figure 2a there appear to be two lines for the LCMVwt titers. Is this a mistake?

We have clarified the legend to Fig. 2 to better indicate that symbols and lines represent individual mice (there are four lines for the LCMVwt group), and that the symbols and lines of the artLCMV group (with undetectable viremia) are overlaid (lines 797-798).

5. In figure 2d the authors show that interferon release is greatest for the artLCMV vector. While this is interesting, it was very surprising that there was no interferon release with the vaccinia virus vector and very low for the rLCMV vector. Were all these viruses given by IV administration? It may be worth showing the vaccinia virus induces interferon by alternate routes of administration, such as subcutaneous or tail vein scratch. These results suggest that the vaccinia is attenuated in some way and perhaps would not be a good control in the other experimental settings in which it has been used.

Following the reviewer's request we have performed subcutaneous and intradermal immunizations with vaccinia virus vectors. The results of the serum IFN-I measurements are reported in the new Supplementary Fig. 2a. Analogously to the intravenous route used in the experiment to Fig. 2d, these alternative routes did not induce any detectable systemic IFN-I either. This is in line with published literature documenting that vaccinia virus fails to induce detectable systemic IFN-I. This lack of systemic IFN-I upon vaccinia virus infection is owed to the fact that vaccinia viruses encode for several independent antagonists of the IFN-I system, which notably includes a decoy receptor besides antagonists of IFN-I induction (Waibler et al. J Virol. 2009;83(4):1563-71, Waibler et al. J Virol. 2007;81(22):12102-10, Symons et al. Cell. 1995;81(4):551-60, Chang et al. Proc Natl Acad Sci U S A. 1992; 89(11):4825-9). Indeed, the rVACC-OVA vector used in our study is derived from the virulent Western Reserve strain for which these antagonists have been described in detail (Waibler et al. J Virol. 2009;83(4):1563-71).

We have amended the respective results section to specify that the lack of type I interferon responses to vaccinia virus immunization was expected based on the known virally encoded IFN-I antagonists (line 162-163).

Reviewer #2 (Remarks to the Author):

In this study the group of Kallert et al report on the development and characterization of a new attenuated, replicating vaccine vector based on molecular modification of the ambisense RNA virus LCMV.

The authors start by demonstrating the make-up of the vector as well as the attenuation and genetic and phenotypic stability of their vector platform. Next the immunogenicity of the new vector platform is compared to that of a few other vaccine platforms using a several different types of analyses. As a mechanistic reason for a better induction of CTL responses, the authors demonstrate that the new vector – unlike the original non-replicating variant – induces IL-33, and that a much reduced T-cell response is found in ST2 deficient mice, lacking the relevant receptor for this cytokine. Then the vector is studied in several relevant tumor model systems, and the superiority of the new vector is clearly established. Finally, they show that the primed antigen-specific CD8 T cells infiltrate the tumor site and create an inflammatory site.

Overall, this is an interesting paper on a novel type of vaccine vector, well-written and based on a very clear and logic strategy. The results are convincing and highly pertinent. From what is presented it is clear that the new vector platform can be applied to induce strong CD8 T-cell responses, albeit I do not believe the authors have been absolutely fair in their selection of vector systems to be used for comparisons, e.g. i.v. injection of adenovectors is not the ideal route and, similarly, the vaccinia vector should perhaps have been given intradermally for a more relevant comparison.

In response to the reviewer's critique we have performed an immunization experiment comparing the following vectors and routes of administration:

- i) artLCMV-OVA intravenously (i.v.)
- ii) rAd-OVA intramuscularly (i.m.)
- iii) rVACC-OVA intradermally (i.d.)
- iv) rVACC-OVA subcutaneously (s.c.)

The results are reported in the new Supplementary Fig. 2b-c, showing that artLCMV-induced OVA-specific CD8 T cell and CTL^{eff} responses were significantly higher than those induced by the rAd or rVACC platforms, irrespective of the latter vectors' route of administration.

The authors claim that vector immunity do not substantially inhibit booster response, and I tend to believe them. However, in order to really prove their point demonstrating absence of vector specific antibodies do not suffice. The real test is to vaccinate a group of mice say twice with a vector directed towards X, and then come back and vaccinate such mice with a vector directed towards Y; by comparing the CTL response against the Y antigen in this group to that in a group of previously unvaccinated mice, you will get a clear picture of how much the vector immunity can interfere with subsequent immunization. Arguably, the capacity to inhibit a secondary/tertiary response might differ from that of reducing a primary response, but this seems to the best way possible, unless an adoptive is taken.

We agree with the reviewer that interference by anti-vector immunity can comprise mechanisms other than neutralizing antibodies and we had no intention to claim that artLCMV homologous prime-boost vaccination was unaffected by anti-vector immunity. After showing i) that neutralizing antibodies were not elicited to a measurable extent (Fig. 2n) and ii) that homologous booster vaccination significantly augmented T cell responses (Fig. 2o), the revised manuscript states clearly: “These observations do not, however, rule out the likely possibility that alternative mechanisms of interference such as T cell immunity to viral backbone epitopes (Schirmbeck Mol Ther. 2008;16(9):1609-16) may compete with and thereby attenuate booster responses to vectorized transgenes.” (lines 190-193).

Finally, in the abstract it very indirectly implied that the effector cells raised by the new vector is somehow unique working in part by turning a cold tumor in a hot one. Personally, I tend to believe that this is more a matter of numbers of CTLs available and attracted, not a difference in quality. Unless, the authors want to make this an issue and do comparative adoptive transfers of matched numbers of effector cell , I suggest the sentence is modified to include the possibility that this phenomenon could be a quantitative issue as much as a qualitative one.

We concur with the reviewer that the respective sentence of the original abstract version could be misinterpreted and thus have rephrased as follows:
“Superior anti-tumor efficacy of artLCMV immunotherapy depends on interleukin-33 signaling, and a massive CTL^{eff} influx converts uninflamed “cold” tumors into “hot” ones.” (lines 34-36)

REVIEWERS' COMMENTS:

Reviewer #1 (Remarks to the Author):

The authors have responded well to the previous critiques, and now include a new experiment (Suppl. Figure 2) demonstrating the absence of IFN with vaccinia virus administered through alternative routes of delivery as suggested by both reviewers. They have also amended several statements in the paper better reflecting the data presented. I think this paper will be of significant interest to many readers.

Reviewer #2 (Remarks to the Author):

The authors have addressed my comments in a satisfactory way, and I have no additional queries

We wish to thank both reviewers for their constructive comments, which have been very helpful to improve our manuscript.